# Prediction of serious complications in patients with pulmonary thromboembolism and solid cancer: Validation of the EPIPHANY Index in a prospective cohort of patients from the PERSEO study

Manuel Sánchez-Cánovas[1]*, Paula Jimenez-Fonseca[2], David Fernández Garay[3], Mónica Cejuela Solís[4], Diego Casado Elía[5], Eva Coma Salvans[6], Irma de la Haba Vacas[6], David Gómez Sánchez[7], Ana Fernández Montés[8], Roberto Morales Giménez[9], Mercedes Biosca Gómez de Tejada[10], Virginia Arrazubi Arrula[7], Silvia Sequero López[11], Remedios Otero Candelera[12], Cristina Sánchez Cendra[13], Marina Justo de la Peña[14], Diana Moreno Muñoz[15], Mayra Orillo Sarmiento[16], Eva Martínez de Castro[17], Ignacio García Escobar[18], Alejandro Bernal Vidal[19], Laura Ortega Moran[20], Andrés J. Muñoz Martín[20], Rodrigo Sánchez Bayona[21], María José Martínez Ortiz[22], Francisco Ayala de la Peña[1], Vicente Vicente[1], Alberto Carmona-Bayonas[1]

1 Hematology and Medical Oncology Department, Hospital Universitario Morales Meseguer, Murcia, Spain, 2 Medical Oncology Department, Hospital Universitario Central de Asturias, ISPA, Oviedo, Spain, 3 Medical Oncology Department, Complejo Hospitalario de Jaén, Jaén, Spain, 4 Medical Oncology Department, Hospital Universitario Insular de Gran Canaria, Las Palmas de Gran Canaria, Spain, 5 Medical Oncology Department, Complejo Hospitalario de Salamanca, Salamanca, Spain, 6 Medical Oncology Department, Institut Català d'Oncologia, Hospital Duran i Reynals, Barcelona, Spain, 7 Medical Oncology Department, Complejo Hospitalario de Navarra, Pamplona, Spain, 8 Medical Oncology Department, Complexo Hospitalario Universitario de Ourense, Ourense, Spain, 9 Medical Oncology Department, Hospital General Universitario de Elche, Elche, Spain, 10 Internal Medicine Department, Hospital Universitario Valle d'Hebron, Barcelona, Spain, 11 Medical Oncology Department, Hospital Universitario San Cecilio, Granada, Spain, 12 Pneumology Department, Hospital Universitario Virgen del Rocío, Sevilla, Spain, 13 Medical Oncology Department, Hospital Universitario de Guadalajara, Madrid, Spain, 14 Medical Oncology Department, Hospital Universitari Son Espases, Palma de Mallorca, Spain, 15 Medical Oncology Department, Fundación Hospital Alcorcón, Madrid, Spain, 16 Medical Oncology Department, Hospital del Mar, Barcelona, Spain, 17 Medical Oncology Department, Hospital Universitario Marqués de Valdecilla, Santander, Spain, 18 Medical Oncology Department, Hospital General Universitario Virgen de las Nieves, Granada, Spain, 19 Medical Oncology Department, Hospital Universitario San Juan de Alicante, Sant Joan d'Alacant, Spain, 20 Medical Oncology Department, Hospital General Universitario Gregorio Marañón, Universidad Complutense, Madrid, Spain, 21 Medical Oncology Department, Hospital Universitario 12 de Octubre, Madrid, Spain, 22 Medical Oncology Department, Hospital General Universitario de Santa Lucía, Cartagena, Spain

* manuelsanchezcanovas@gmail.com

## Abstract

### Introduction

There is currently no validated score capable of classifying cancer-associated pulmonary embolism (PE) in its full spectrum of severity. This study has validated the EPIPHANY Index, a new tool to predict serious complications in cancer patients with suspected or unsuspected PE.

**Data Availability Statement:** All relevant data are within the paper and its Supporting information files.

**Funding:** The authors received no specific funding for this work.

**Competing interests:** The authors have declared that no competing interests exist.

## Method

The PERSEO Study prospectively recruited individuals with PE and active cancer or receiving antineoplastic therapy from 22 Spanish hospitals. The estimation of the relative frequency θ of complications based on the EPIPHANY Index categories was made using the Bayesian alternative for the binomial test.

## Results

A total of 900 patients, who were diagnosed with PE between October 2017 and January 2020, were enrolled. The rate of serious complications at 15 days was 11.8%, 95% highest density interval [HDI], 9.8–14.1%. Of the EPIPHANY low-risk patients, 2.4% (95% HDI, 0.8–4.6%) had serious complications, as did 5.5% (95% HDI, 2.9–8.7%) of the moderate-risk participants and 21.0% (95% HDI, 17.0–24.0%) of those with high-risk episodes. The EPIPHANY Index was associated with overall survival (OS) in patients with different risk levels: median OS was 16.5, 14.4, and 4.4 months for those at low, intermediate, and high risk, respectively. Both the EPIPHANY Index and the Hestia criteria exhibited greater negative predictive value and a lower negative likelihood ratio than the remaining models. The incidence of bleeding at 6 months was 6.2% (95% HDI, 2.9–9.5%) in low/moderate-risk vs 12.7% (95% HDI, 10.1–15.4%) in high-risk (p-value = 0.037) episodes. Of the outpatients, serious complications at 15 days were recorded in 2.1% (95% HDI, 0.7–4.0%) of the cases with EPIPHANY low/intermediate-risk vs 5.3% (95% HDI, 1.7–11.8%) in high-risk cases.

## Conclusion

We have validated the EPIPHANY Index in patients with incidental or symptomatic cancer-related PE. This model can contribute to standardize decision-making in a scenario lacking quality evidence.

## Introduction

Pulmonary embolism (PE) has classically been deemed one of the most common and dire complications in patients with cancer [1]. Studies show that the 30-day mortality rates for acute PE, which often requires intensive hospital management, range from 15% to 25% across different series [2–8]. In contrast, in recent years, computed tomography pulmonary angiogram (CTPA) and multidetector CT have enhanced the accuracy of PE diagnoses [9, 10], with increased detection of unsuspected events. The upsurge of this modality has utterly changed the prognostic scenario of some of these PEs, which is reflected in the epidemiological panorama. In a Danish populational registry, 16% of the cases had cancer and, while incidence of PE increased from 2004 to 2014, the episodes detected were gradually less lethal [11]. More specifically in the oncological population, the pooled frequency of incidental events discovered during cancer staging was 3.2% in a recent meta-analysis [12], exceeding the estimation made one decade earlier [13]. With these data, incidental PE accounts for more than 50% of all episodes of cancer-related PE [10, 14]. Given that the prevalence of PE in autopsy series is calculated to be 15–30% [15], incidental diagnoses can be expected to increase, as technology improves.

Ironically, this trend affects decision making in the oncologic population, inasmuch as there is currently no validated method of classification that facilitates choosing episodes with a

good prognosis for ambulatory management [3, 8, 16, 17]. Inarguably, averting hospitalizations would lower costs, avoid iatrogenesis, and improve quality of life [18]. Although there are some limited data and guidelines suggesting that outpatient management of cancer-related PE is feasible, high-quality prospective data supporting this approach is still lacking [3, 19–21]. To date, clinical trials of home management have selected participants using decision rules, such as the Hestia criteria or adaptations carried out by other authors [3, 19], contraindicating lowering the level of support in patients with any exclusion criterion [19, 22, 23]. These exclusion criteria are based on the combination of altered vital signs and factors that point toward a high risk of bleeding or other contraindications to receiving treatment in the home. Nevertheless, there are pockets of uncertainty, given the oncological population's greater risk of rethrombosis or major bleeding [24, 25], the greater multifariousness of PE, and scant experience [3, 19, 26].

Presently, there is no validated score capable of classifying unsuspected, cancer-related PE. Surprisingly, the scores developed specifically for individuals with cancer (e.g., RIETE or POMPE-C) are not suitable for stratifying incidental PE nor do they predict key outcomes, such as bleeding [4, 5, 8]. Meanwhile, validated scores for acute, symptomatic PE in the general population (e.g., simplified PESI) fail to capture the heterogeneity of the patient with cancer [8, 25, 27–29].

The EPIPHANY Index was designed as a modified extension of the Hestia clinical decision rule to pragmatically select patients with low-risk, cancer-related PE [14]. In contrast with other proposals, the EPIPHANY Index is applicable across the entire gamut of PE severity (suspected or unsuspected). A meta-analysis comparing the accuracy of clinical decision rules found that higher sensitivity is obtained with the Hestia criteria and the EPIPHANY Index [30]; however, validations of both sets of criteria in patients with cancer are meager [31].

Against this background, the PERSEO ("*Pulmonary Embolism Risk Stratification & End results in Oncology*") Study pursues the prospective, multicenter validation of the EPIPHANY Index and Hestia criteria in a broad sample of patients with cancer, with a view to their possible appropriateness in decision making.

## Method

### Patients and study design

PERSEO prospectively and consecutively recruited patients with PE and cancer from 22 Spanish tertiary hospitals. Patient recruitment was conducted during the acute phase of PE (within the first three hours) to enable the direct collection of vital signs and symptom data. Eligibility criteria comprised adults ($\geq$18 years of age) with an active solid neoplasm or anti-tumor treatment at the time of the thrombotic event or in the previous 30 days. All the participants had to have a diagnosis of PE confirmed by an objective imaging technique (CTPA, high probability scintigraphy, or computed tomography scheduled to assess tumor response or for other reasons). In the case of multiple events, only one episode of PE per subject (the first one) was allowed. Subjects were recruited by medical oncologists in outpatient clinics, the Emergency Department, or during hospitalization. No recommendation was made regarding management on the basis of any given clinical decision rule or specific treatment approach, and the cases were managed as per regular practice. The investigators were not blind to outcomes, albeit nor were they aware of the EPIPHANY Index during management. To detect serious complications, all patients still alive were followed for a minimum of 30 days. In the case of participants treated on an ambulatory basis, follow up was conducted by phone and in outpatient clinics.

The study was approved by the Ethics Committees of all the participating centers and was conducted in accordance with the requirements put forth in the international guidelines regarding epidemiological studies [32], as well as the Declaration of Helsinki and its subsequent revisions. All participants signed an informed consent form.

## Objective

The main objective was to validate the EPIPHANY Index and Hestia criteria in subjects with cancer. The secondary objectives of this study were (1) to compare predictive parameters with other models, including Hestia criteria, RIETE model, PESI, simplified PESI, Spanish score, and Geneva score [5, 22, 25, 28, 29, 33]; (2) assessing the prognostic effect of PE presentation; (3) analyzing bleeding events, rethrombosis, and mortality, and studying the evolution of participants treated on an ambulatory basis.

## Variables

In our model validation process, we selected serious complications within 15 days as the primary endpoint. This choice was made based on its better fit with the clinical scenario of managing PE in cancer patients as outpatients. This decision was previously supported in the literature [8, 14]. This endpoint includes the development of any of the following events: hypotension (systolic blood pressure <90 mmHg), acute respiratory insufficiency, fibrinolysis, major bleeding (intracranial, intraspinal, intraocular, retroperitoneal, or pericardial, associated with decreased hemoglobin by at least 2 g/dl or requiring the transfusion of two units of red blood cells), right ventricle dysfunction [34], acute kidney failure, admission into the Intensive Care Unit, need of cardiopulmonary resuscitation, non-invasive mechanical ventilation, orotracheal intubation, sepsis, death, or any other event the investigator deems serious. These categories were not mutually exclusive. To be deemed outcomes, these events must have occurred after the objective diagnosis of PE. Should they have occurred before then (e.g., at the time of debut), the events were coded as predictors. The reason for this approach was to rapidly classify patients based on the initial Hestia criteria and specifically investigate their ability to predict the eventual development of severe complications after the objective diagnosis of PE, based on the baseline clinical presentation.

The Hestia criteria [22] (systolic blood pressure <100mmHg, arterial oxygen saturation <90%, respiratory rate ≥30 breaths per minute, pulse ≥110 beats per minute, sudden or progressive dyspnea, other serious complications, constituting admission criteria in and of themselves) were adapted to the oncological population, including: clinically relevant bleeding, high risk of bleeding, or platelets <50,000mm$^{-3}$. The other five models—Hestia criteria, RIETE model, PESI, simplified PESI, Spanish model, and Geneva score—were evaluated based on their original descriptions and validated cut-off points [5, 25, 28, 29, 33]. Each prognostic category was assigned a 30-day mortality probability value. Predicted and observed deaths at 30 days were compared to assess the statistical validity of the models.

The EPIPHANY Index [35] (Fig 1) incorporates these criteria together with other particular characteristics of individuals with cancer, such as the Eastern Cooperative Group Performance Status (ECOG-PS), evaluation of tumor response prior to or during the study of the PE using RECIST v1.1 criteria, previous primary tumor resection, oxygen saturation, and the presence or absence of PE-specific symptoms (acute or progressive dyspnea, chest pain, or syncope). The EPIPHANY index is accessible via the web: https://www.prognostictools.es/epiphany/inicio.aspx. Patient and cancer baseline characteristics were attained from the clinical history and from the interview at the time of the PE. Missing for these predictors were not allowed.

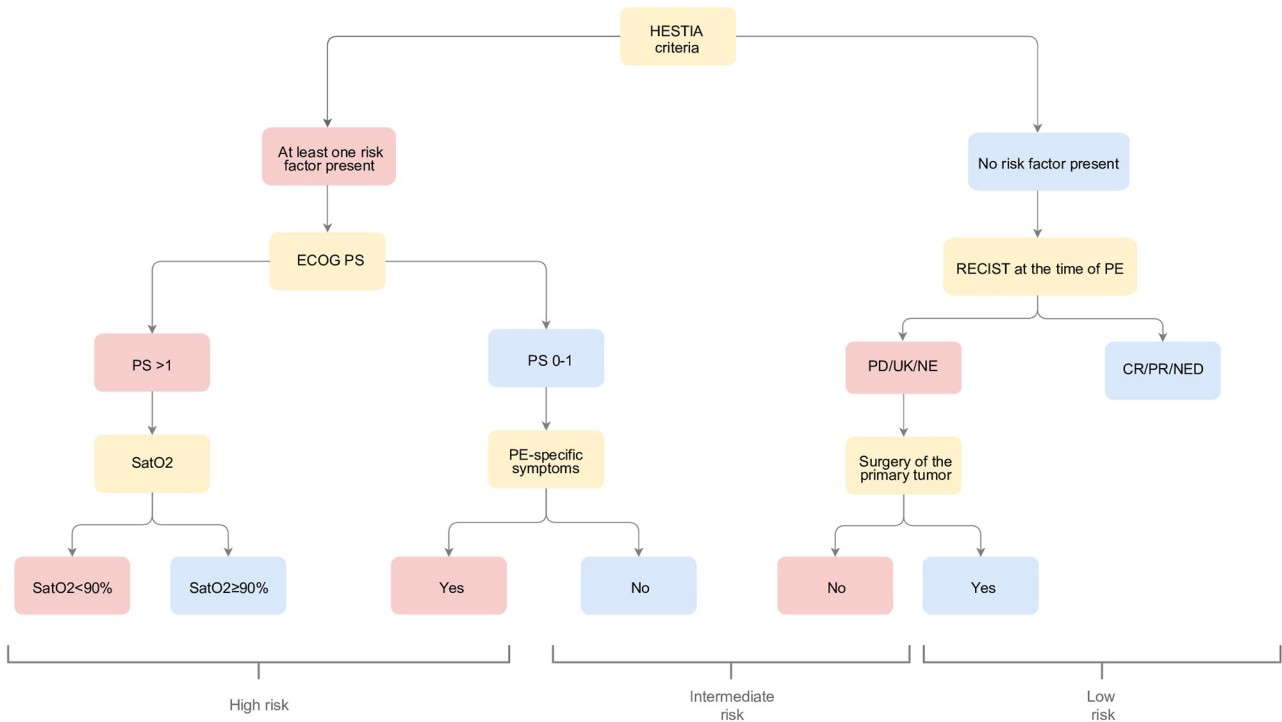

**Fig 1. The EPIPHANY Index.** Abbreviations: ECOG PS, Eastern Cooperative Group Performance status; SpO2, oxygen saturation; PE, pulmonary embolism; RECIST, Response Evaluation Criteria In Solid Tumors; PD, progressive disease; UK, unknown; NE, not evaluable, CR, complete response; PR, partial response; NED, no evidence of disease. The HESTIA criteria are typified by the presence of at least one of the following: systolic blood pressure <100 mm Hg, arterial oxygen saturation <90%, respiratory rate ≥30 breaths per minute, pulse ≥110 beats per minute, sudden or progressive dyspnea, other serious complications, constituting admission criteria in and of themselves, and clinically relevant bleeding, high risk of bleeding, or platelets <50 000 mm−3. These criteria should be assessed immediately prior to the time of radiological diagnosis of PE.

Overall survival (OS) was estimated from the date of the PE until demise due to any cause, censoring all subjects without any event at last follow up.

## Statistics

The estimation of relative frequency $\theta$ of serious complications based on the EPIPHANY Index categories was performed by means of the Bayesian alternative for the binomial test, applying Jeffreys non-informative prior [36]. The sample size was calculated to estimate $\theta$ in the low-risk EPIPHANY category with a desired degree of precision in the posterior distribution ($\theta$ = 0.02; 95% highest density interval [HDI] width <0.02 at least 90% of the time) [36]. Assuming approximately one-third low-risk subjects, a sample size of 900 participants was estimated.

As measures of performance, sensitivity, specificity, positive and negative predictive values (PPV and NPV), and positive and negative likelihood ratios (PLR and NLR) were estimated. The NPV is the probability of having no complications when the subject is not identified as high risk. Causal inference as to the contribution of the individual variables was explored via frequentist multivariable logistic regression, including the 6 variables of the EPIPHANY index. Discrimination was appraised by comparing the area under the curve-receiver operating characteristics curve (AUC-ROC) using the DeLong test. OS was estimated using the Kaplan-Meier method. In the presence of competing risks, the cumulative incidence of rethrombosis and bleeding events was probed with the Aalen-Johansen estimator. The time to event functions

were compared using Gray or log-rank tests in each case. Analyses were carried out using the R v4.05 software including the cmprsk, survival, pROC, and rjags libraries [37–40].

## Results

### Patients

The sample consists of 900 patients treated between October 2017 and January 2020. Baseline characteristics can be found in Table 1. The median age was 66 years (range 21–94), with a predominance of males (57.4%, n = 517), and good functional status (ECOG-PS 0–1, 64.6%, n = 582). The most frequent tumors were lung (26.3%, n = 237), colorectal (18.6%, n = 168), and breast cancer (8.1%, n = 73); most were stage IV (78.3%, n = 705). Seven hundred participants (77.7%) were in active treatment, the most frequent anticancer therapy being chemotherapy (60.6%, n = 546). The primary tumor had not been resected in two thirds (66.5%, n = 599). Almost half (48.6%, n = 437) of the episodes were unsuspected, asymptomatic. Full-dose low-molecular-weight heparin was the treatment administered in 92.9% (n = 836). The baseline characteristics stratified by episode subtypes are shown in S1 Table.

### Outcomes

Serious complications within 15 days were confirmed in 107 patients (11.8%, 95% CI, 9.8–14.1). The median time from PE diagnosis until the appearance of said complications was 4 days (range, 0–15). Table 2 comprises a list of the serious complications, the most common being acute respiratory insufficiency (6.6%, n = 59), major bleeding (2.6%, n = 23), and shock (2.3%, n = 21). The incidence of complications was comparable among the entire cohort, including individuals with and without previous anticoagulation therapy, with complication rates of 107 (11.8%), 93 (11.8%), and 14 (12.6%), respectively.

At the time of analysis, 434 events of death (48.2%) were detected after a median follow up in patients still alive of 11.2 months (95% confidence interval [CI], 10.2–12.2). Median OS was 10.0 months (95% CI, 7.9–11.9). The causes of 30-day mortality are listed in S2 Table, the most prevalent being tumor progression. No deaths were due to rethrombosis in the first 30 days following PE diagnosis, albeit 6.7% of the expiries during this period were attributable to major bleeding.

### Validation of the EPIPHANY Index

The reconstruction of the decision tree is displayed in S1 Fig
. Serious complications at 15 days arose in 2.4% of the patients with episodes classified as low-risk (95% HDI, 0.8–4.6%), in 5.5% (95% HDI, 2.9–8.7) of those with moderate-risk episodes, and in 21.0% (95% HDI, 17.0–24.0) of subjects with high-risk episodes (posterior probability $\theta_{moderate-risk} > \theta_{low-risk} = 95.8\%$, posterior probability $\theta_{high-risk} > \theta_{moderate-risk} = 100\%$) (Table 3). No substantial changes were observed in the performance of the index after excluding individuals who had received anticoagulants before the PE, as shown in S1B Fig.

The Bayesian alternative for the binomial test is shown in Fig 2. The EPIPHANY Index correlated consistently with OS (Fig 3). Applying only the Hestia criteria, the 15-day rate of serious complications was 18% (95% HDI, 15–21%) in the presence of at least one factor vs 2.5% (95% HDI, 1.0–4.2) in the absence of all criteria (posterior probability $\theta+_{Hestia} > \theta-_{Hestia} = 100\%$).

The dichotomized EPIPHANY Index (high vs other risks) is associated with a NPV of 97.8% (95% CI, 95.0–99.2) and NLR of 0.16 (95% CI, 0.07–0.39) (S3 Table). In episodes of suspected PE, the EPIPHANY Index is associated with a NPV of 90.9% (95% CI, 62.2–98.3) and NLR of 0.38. As for 30-day mortality, the EPIPHANY Index exhibits a NPV of 99.1% (95% CI,

**Table 1. Baseline characteristics of the sample.**

| Baseline characteristics | N = 900 (100%) |
|---|---|
| **Sex, male** | 517 (57.4) |
| **Age, median (range)** | 66 (21–94) |
| **ECOG-PS** | |
| *0* | 136 (15.1) |
| *1* | 446 (49.6) |
| *2* | 229 (25.4) |
| *3* | 82 (9.1) |
| *4* | 7 (0.8) |
| **Most common tumors** | |
| *Lung* | 237 (26.3) |
| *Colorectal* | 168 (18.7) |
| *Breast* | 73 (8.1) |
| *Pancreatic* | 64 (7.1) |
| *Bladder* | 47 (5.2) |
| *Stomach* | 45 (5.0) |
| *Ovarian* | 40 (4.4) |
| *Central nervous system* | 31 (3.4) |
| *Endometrial* | 25 (2.8) |
| **TNM classification, stage IV** | 705 (78.3) |
| **RECIST at PE diagnosis** | |
| *Not evaluable* | 418 (46.4) |
| *Progression* | 204 (22.7) |
| *Stable disease* | 144 (16) |
| *Partial response* | 94 (10.4) |
| *Complete response/ NED* | 40 (4.4) |
| **Comorbidities** | |
| *Chronic cardiovascular disease* | 90 (10.0) |
| *COPD* | 79 (8.8) |
| *Chronic kidney disease* | 34 (3.7) |
| *Chronic liver disease* | 15 (1.4) |
| *Major surgery in the previous 3 months* | 50 (5.6) |
| **Smoking status** | |
| *Non-smoker* | 160 (17.8) |
| *Ex-smoker* | 31 (3.4) |
| *Active smoker* | 312 (34.7) |
| *Unknown* | 397 (44.1) |
| **Previous thrombosis** | 139 (15.5) |
| **Anticoagulation therapy prior to the diagnosis of PE** | 110 (12.3) |
| **Anti-aggregation therapy at diagnosis of PE** | 71 (7.9) |
| **PE diagnostic technique** | |
| *Angio-CT* | 301 (33.4) |
| *Ventilation/ perfusion scintigraphy* | 18 (2.0) |
| *CT for response assessment* | 437 (48.6) |
| *CT for other reasons* | 144 (16) |
| **Type of PE** | |
| *Unilateral* | 502 (55.8) |
| *Bilateral* | 398 (44.2) |

(*Continued*)

**Table 1.** (Continued)

| Baseline characteristics | N = 900 (100%) |
|---|---|
| *Central* | 209 (23.2) |
| *Peripheral* | 333 (37.0) |
| *Central and peripheral* | 358 (39.8) |
| *Suspected* | 319 (35.4) |
| *Unsuspected, symptomatic* | 144 (16.0) |
| *Unsuspected, asymptomatic* | 437 (48.6) |
| **Type of care** | |
| *Outpatient* | 351 (39.0%) |
| *Inpatient* | 549 (61.0%) |

Abbreviations: COPD, Chronic Obstructive Pulmonary Disease; CT, computed tomography; ECOG-PS, Eastern Cooperative Oncology Group; N, number of patients; NED, No Evidence of Disease; PE, Pulmonary Embolism; RECIST, Response Evaluation Criteria In Solid Tumors.

96.9–99.8) and NLR of 0.06 (95% CI, 0.01–0.26) (S4 Table). Against other scores, the EPIPH-ANY index has a greater NPV to rule out complications in symptomatic PE. Both exhibited good discriminatory capacity in unsuspected PE (Table 3).

Different data splits based on geographic distribution did not reveal substantial changes with respect to overall classification performance (S2 Fig).

**Table 2. Severe 15-day complications grouped by treatment site.**

| Complication* | Patients in outpatient care | Patients in inpatient care | Total |
|---|---|---|---|
| | N = 351 (100%) | N = 549 (100%) | N = 900 (100%) |
| **Acute respiratory insufficiency** | 4 (1.1) | 55 (10) | 59 (6.6) |
| **Major bleeding** | 3 (0.9) | 20 (3.6) | 23 (2.6) |
| **Hypotension** | 0 | 21 (3.8) | 21 (2.3) |
| **(SBP <90 mmHg)** | 1 (0.3) | 18 (3.3) | 19 (2.1) |
| **Sepsis** | 0 | 18 (3.3) | 18 (2) |
| **ICU admission** | 0 | 16 (2.9) | 16 (1.8) |
| **RV dysfunction** | 0 | 11 (2%) | 11 (1.2) |
| **Acute kidney failure** | 0 | 7 (1.3%) | 7 (0.8) |
| **NIV** | 0 | 6 (1.1%) | 6 (0.7) |
| **OTI** | 0 | 5 (0.9%) | 5 (0.6) |
| **Fibrinolysis** | 0 | 1 (0.2%) | 1 (0.1) |
| **Need for CPR maneuvers** | 4 (1.1) | 61 (11.1) | 65 (7.2) |
| **Death** | 1 (0.3) | 17 (3.1%) | 18 (2) |
| **Other complications ‡** | 9 (2.6) | 98 (17.9) | 107 (11.9) |

Abbreviations: CPR, cardiopulmonary resuscitation; ICU, Intensive Care Unit; N, number of patients; NIV, non-invasive mechanical ventilation; OTI, orotracheal intubation; RV, right ventricle; SBP, Systolic Blood Pressure.

*These events are not mutually exclusive.

‡ In the category "other complications", the following serious complications were recorded: acute intestinal ischemia (n = 1), need for thrombectomy (n = 1), severe pain requiring intravenous opioids (n = 1), severe pleural effusion requiring thoracentesis (n = 1), severe anemia (Hb 6 mg/dL) (n = 1), pulmonary infarction and bronchopleural fistula (n = 1), intestinal perforation (n = 1), acute pancreatitis (n = 1), grade 4 thrombocytopenia due to chemotherapy (n = 1), severe pulmonary hypertension (n = 1), ischemic stroke (n = 2), bronchoaspiration (n = 1), hemothorax (n = 1), cardiac tamponade (n = 1), severe infection (n = 2), and other serious complications, not specified (n = 1).

**Table 3. Distribution of risk classes and 15-day serious complications according to the EPIHANY index/ modified Hestia criteria.**

| Category | Complications, n/N | Complications, % (95% HDI) | Mortality, n/N | Mortality, % (95% HDI) |
|---|---|---|---|---|
| **Low-risk Epiphany** | 5/ 232 | 2.2 (0.8–4.6) | 1/232 | 0.5 (0.0–2.0) |
| **Intermediate-risk Epiphany** | 12/ 229 | 5.5 (2.9–8.7) | 7/229 | 3.1 (1.3–5.9) |
| **High-risk Epiphany** | 90/ 439 | 20.5 (16.9–24.4) | 57/439 | 13.0 (10.0–16.4) |
| **Low-risk Hestia** | 8/ 350 | 2.3 (1.0–4.2) | 5/ 350 | 1.4 (0.5–3.1) |
| **High-risk Hestia** | 99/ 550 | 18.0 (14.9–21.4) | 60/ 550 | 10.9 (8.5–13.7) |

Abbreviation: HDI, highest density interval, n, number with complications; N, number of patients.

The cumulative incidence of venous rethrombosis at 6 months as per the EPIPHANY Index was 2.9% (95% CI, 0.6–5.3) in low/moderate-risk vs 7.0% (95% CI, 4.9–9.1) in high-risk episodes (p-value = 0.651, Gray's test) (S3 Fig). The incidence of bleeding (any grade) at 6 months was 6.2% (95% CI, 2.9–9.5) in low/moderate-risk vs 12.7% (95% CI, 10.1–15.4) in high-risk episodes (Gray's test, p-value = 0.037) (S4 Fig).

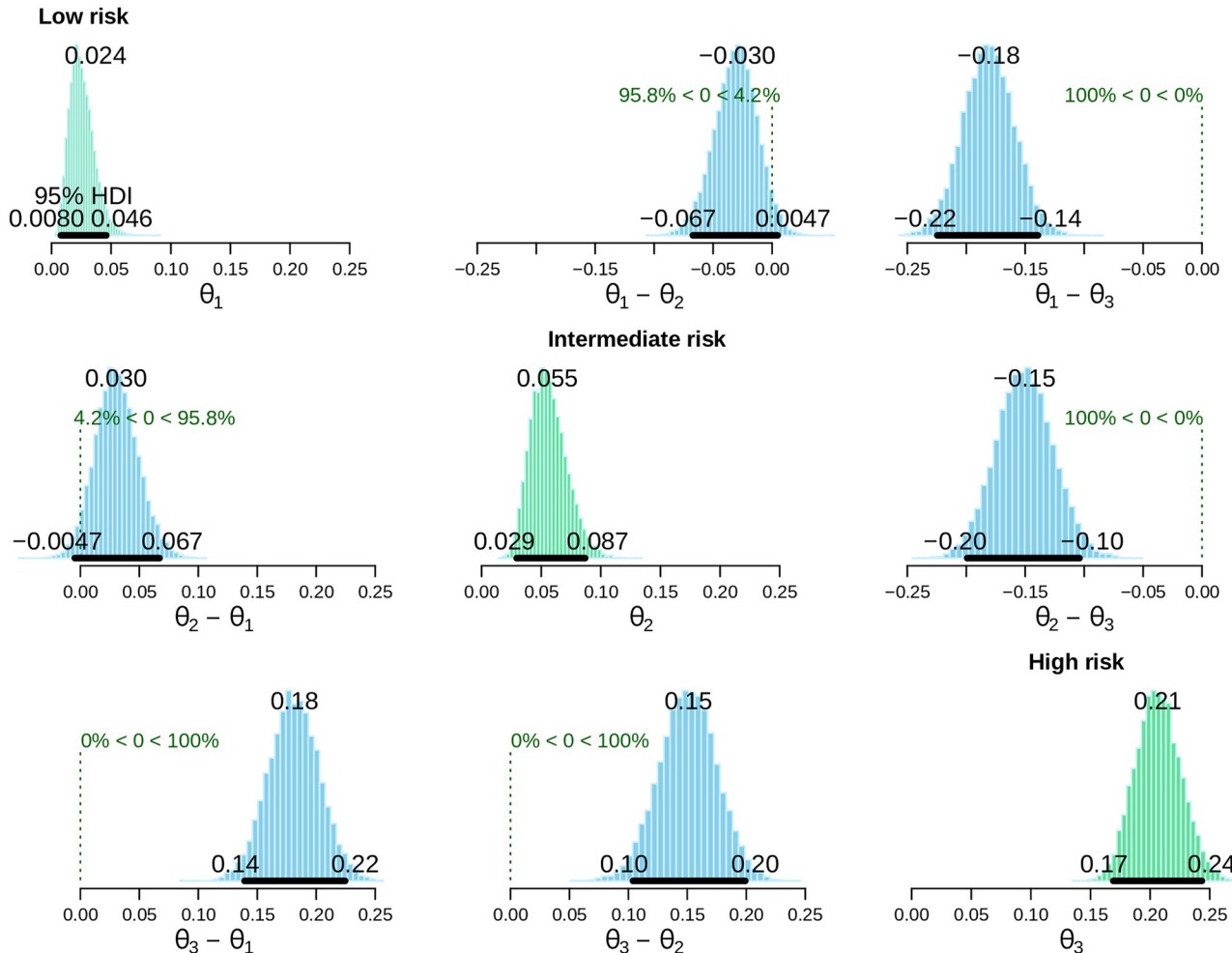

**Fig 2. Bayesian alternative for the binomial test.** The endpoint is the 15-day complication rate, represented by the θ parameter, the subscript representing the risk category ($\theta_1$ low risk, $\theta_2$ intermediate risk, $\theta_3$ high risk). The green panels characterize the posterior distribution of θ for each category. The blue panels display the posterior for the difference in proportions. In each case, the probability that the difference is greater or less than zero is reported. Abbreviations: HDI, highest density interval.

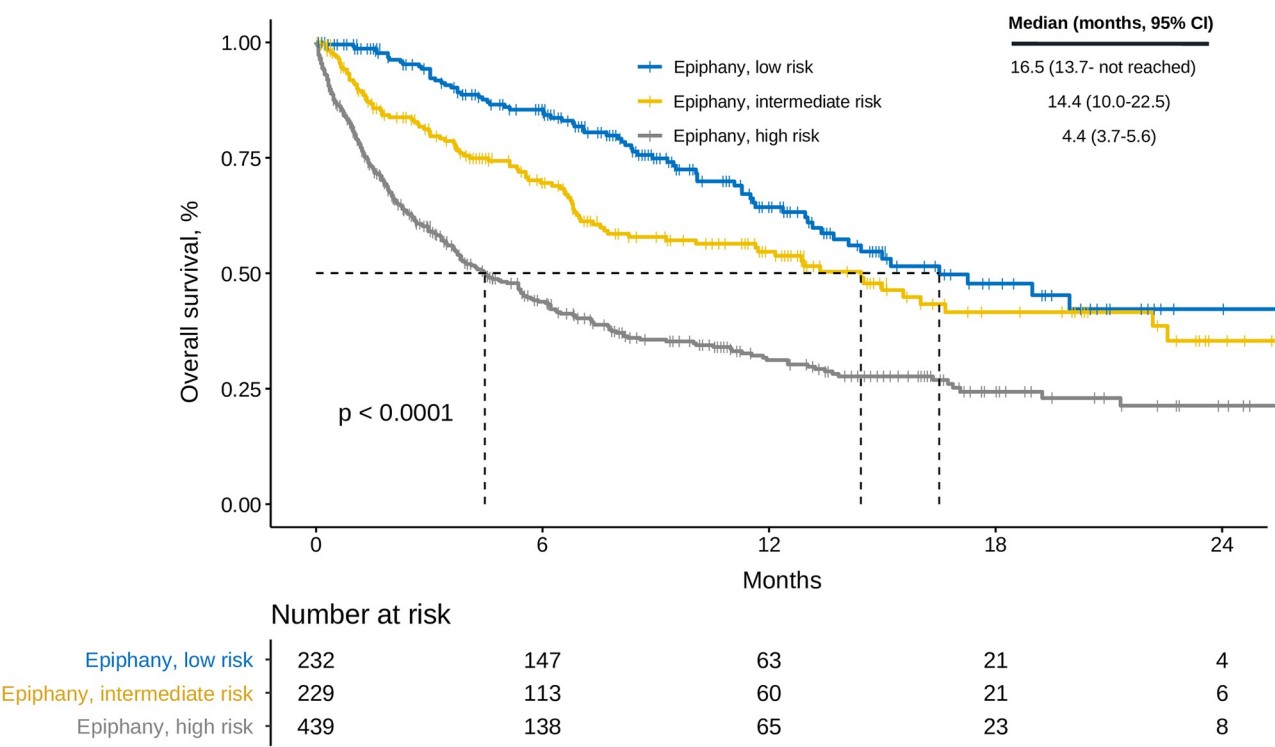

**Fig 3. Overall survival based on the EPIPHANY Index categories.** Abbreviations: CI, confidence interval.

In the logistic regression for serious complications, the covariates having the greatest weight were the modified Hestia criteria, odds ratio (OR) 3.6 (95% CI, 1.5–8.4), and disease in progression or not evaluable, OR of 2.1 (95% CI, 1.1–4.0) (S5 Table).

## Performance in ambulatory patients

The 15-day rate of serious complications in ambulatory patients was 2.5% (95% HDI, 1.1–4.8). These complications were detected later in outpatients vs inpatients (median 9 vs 3 days, Wilcoxon's test, p-value = 0.025). None of the subjects managed in their homes died as a result of acute PE complications. Complications at 15 days were recorded in 5/273 subjects with low/intermediate EPIPHANY risk (2.1%, 95% HDI, 0.7–4.0) vs 4/78 (5.9%, 95% HDI, 1.7–11.9) in high EPIPHANY risk (posterior probability $\theta_{high} > \theta_{moderate/low}$ = 94.9%) (S6 Table and S5 Fig). S7 Table reflects the 30-day mortality rates. 4 patients (1.1%) treated as outpatients died during this period (2 cases were attributed to progression, 2 to mixed cause). The distribution of the modified Hestia criteria in ambulatory patients is displayed in S8 Table.

## Discussion

This study has validated the EPIHANY Index, a new tool to predict serious complications in patients with PE and cancer, with potential usefulness for outpatient management decisions. This index is an extension of the Hestia criteria, developed to increase its discriminatory power in the oncological population by including specific variables, such as ECOG-PS, tumor response as per RECIST, surgery on the primary tumor, or symptoms [8, 14, 31]. The Hestia criteria have been used here in an adapted form to capture the higher risk of bleeding in this population [10, 22] Our study confirms the validity of these criteria and their extension in the

EPIPHANY Index, which endorses the use of sets of exclusion criteria similar to those proposed by other authors over sum scores [8, 23, 30]. Both are more reliable than other scales to rule out complications, inasmuch as they are capable of discriminating between incidental PEs not classifiable by other models.

Certain nuances must be taken into account when interpreting these results. First, the EPIPHANY Index predicts the 15-day complication rate, since this endpoint was considered to be the most relevant when making decisions about ambulatory management [8]. In contrast, the other scores predict 30-day all-cause mortality. This latter endpoint involves the risk of incurring in clinical contradictions, such as overtreating high-risk patients receiving palliative care, given that most predicted events will be attributable to tumor progression with a marginal role of the PE [14]. Second, the probability that the outpatient actually cares about is the NPV, and this parameter is higher in the EPIPHANY Index and Hestia criteria vs the other models. In exchange, all the models exhibited a low PPV, especially in scenarios that are *a priori* low-risk, such as incidental PE. This limitation is shared by other scores used in support care, generally more useful to rule out serious complications than to identify them [41]. By way of example, high-risk subjects as per RIETE [5] had a 22% complication rate in this sample, similar to that seen with Hestia/EPIPHANY high-risk individuals. Undoubtedly, the asymmetric consequences of each error (e.g., malapropos scaling back vs intensifying therapy) must be factored into any treatment algorithm. The most nefarious result is that subjects develop complications after being mistakenly classified as low-risk. In contrast, if we persevere with the pragmatic approach, the low PPV poses a tolerable risk. In fact, hospitalization founded on Hestia criteria is hardly negligible in practical terms, even in the absence of a true basis of severity. In any case, it is possible to envision predictors with a greater PPV, albeit disconnected from a utilitarian orientation, rendering their practicality questionable. Third, the capacity to discriminate prognosis in incidental episodes is one of the most noteworthy contributions, since the rest of the scores were designed for acute, symptomatic PE [42] and do not contain the variables that make it possible to discriminate levels of risk within unsuspected PE. Our results are consistent with the retrospective validation by Ahn et al, focused on incidental PE, that reported a 15-day complication rate of 3.4, 8.9, and 23.8% for low-, intermediate-, and high-risk EPIPHANY episodes [31]. In return, the EPIPHANY Index/Hestia criteria perform worse in acute, symptomatic PE, as patients in this scenario are rarely classified as low risk. Nevertheless, the NLR of EPIPHANY index/Hestia criteria applied to symptomatic PE is comparable to the remaining validated scales, reflecting the fact that operationally, they perform on a par. Finally, the EPIPHANY Index can discriminate bleeding risk, making it appealing in the oncologic population, as it captures multiple variables associated with bleeding that is the most serious iatrogenic complication.

It is important to acknowledge the potential limitations of our study, specifically the possible impact of prior anticoagulant use on the performance of the model. This is especially relevant in cases where supratherapeutic doses of heparin need to be administered for rethrombosis. As such, we caution readers to consider this variable when interpreting our results and suggest that further research is necessary to determine the true impact of this variable on the performance of the model. Another limitation of the study is that its observational design did not intend to consider any intervention based on EPIPHANY and, in fact, the investigators were not explicitly aware of the classification. Yet, the subjects receiving ambulatory treatment in this series tended to *de facto* uphold most of the Hestia criteria, the most tolerated factors being the risk of bleeding (16%) and dyspnea (13%). Validation in clinical trials is needed. In any case, the favorable results of this empirical selection are similar to those reported with the most refined models [19, 22]. The absence of a concordance analysis and the unblinded evaluation of outcomes are other limitations [43]. Technically, another limitation is

the frailty of the classifiers based on decision trees, with interactions that are often not validated or vary across populations. This method was applied to adapt to decision-making algorithms in the real world [14]. It is certainly feasible that other, more complex methods based on ensembles (e.g., random forest, gradient boosting), might improve the PPV. Another limitation is that only Spaniards have been included. Caution is advised when extrapolating to other populations, although, a priori, we would not expect large differences to emerge.

Ultimately, in this prospective, multicenter study, we have validated the EPIPHANY Index/ Hestia criteria in patients with acute, cancer-related PE. This can help to pave the way to standardize decision-making in a scenario in which there was no quality evidence. Our results are relevant to supporting ambulatory management of patients with low-risk PE, particularly in the case of unsuspected events, inasmuch as they already have a lower risk of complications.

## Supporting information

**S1 File.**
(XLSX)

**S1 Fig.**
(TIF)

**S2 Fig.**
(TIF)

**S3 Fig.**
(TIFF)

**S4 Fig.**
(TIFF)

**S5 Fig.**
(TIF)

**S1 Table. Baseline characteristics of the sample stratified by the laterality of pulmonary embolism.**
(DOCX)

**S2 Table. 30-day causes of death.**
(DOCX)

**S3 Table. Comparison of performance of various prediction models (serious complications).**
(DOCX)

**S4 Table. Comparison of performance of various prediction models (30-day mortality).**
(DOCX)

**S5 Table. Multivariate analysis: Predictors of serious complications.**
(DOCX)

**S6 Table. Serious complications at 15 days based on type of PE & location of management.**
(DOCX)

**S7 Table. Distribution of risk classes and 30-day mortality by EPIHANY index/ modified HESTIA criteria and treatment site.**
(DOCX)

**S8 Table. HESTIA criteria in patients cared for in hospital and at home.**
(DOCX)

## Acknowledgments

### Ethics approval

The study was approved by the Research Ethics Committee of the Hospital General Universitario José María Morales Meseguer (code: C.P. PERSEO—C.I. EST: 57/17, 26 October 2017) and by the Spanish Agency of Medicines and Medical Devices (AEMPS) (6 October 2017). The study has been conducted in accordance with the ethical standards of the 1964 Declaration of Helsinki and its later amendments. This study is an observational, non-interventionist trial.

## Author Contributions

**Formal analysis:** Manuel Sánchez-Cánovas, Alberto Carmona-Bayonas.

**Investigation:** Manuel Sánchez-Cánovas, Paula Jimenez-Fonseca, David Fernández Garay, Mónica Cejuela Solís, Diego Casado Elía, Eva Coma Salvans, Irma de la Haba Vacas, David Gómez Sánchez, Ana Fernández Montés, Roberto Morales Giménez, Mercedes Biosca Gómez de Tejada, Virginia Arrazubi Arrula, Silvia Sequero López, Remedios Otero Candelera, Cristina Sánchez Cendra, Marina Justo de la Peña, Diana Moreno Muñoz, Mayra Orillo Sarmiento, Eva Martínez de Castro, Ignacio García Escobar, Alejandro Bernal Vidal, Laura Ortega Moran, Andrés J. Muñoz Martín, Rodrigo Sánchez Bayona, María José Martínez Ortiz, Francisco Ayala de la Peña, Vicente Vicente, Alberto Carmona-Bayonas.

**Methodology:** Manuel Sánchez-Cánovas, Paula Jimenez-Fonseca, Alberto Carmona-Bayonas.

**Writing – original draft:** Manuel Sánchez-Cánovas, Paula Jimenez-Fonseca, Alberto Carmona-Bayonas.

**Writing – review & editing:** David Fernández Garay, Mónica Cejuela Solís, Diego Casado Elía, Eva Coma Salvans, Irma de la Haba Vacas, David Gómez Sánchez, Ana Fernández Montés, Roberto Morales Giménez, Mercedes Biosca Gómez de Tejada, Virginia Arrazubi Arrula, Silvia Sequero López, Remedios Otero Candelera, Cristina Sánchez Cendra, Marina Justo de la Peña, Diana Moreno Muñoz, Mayra Orillo Sarmiento, Eva Martínez de Castro, Ignacio García Escobar, Alejandro Bernal Vidal, Laura Ortega Moran, Andrés J. Muñoz Martín, Rodrigo Sánchez Bayona, María José Martínez Ortiz, Francisco Ayala de la Peña, Vicente Vicente.

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
