## [Decision Letter · Decision Letter 0]

16 Feb 2023

PONE-D-22-07801Prediction of serious complications in patients with cancer and pulmonary thromboembolism: validation of the EPIPHANY Index in a prospective cohort of patients from the PERSEO StudyPLOS ONE

Dear Dr. Sánchez Cánovas, 

Thank you for submitting your manuscript to PLOS ONE. After careful consideration, we feel that it has merit but does not fully meet PLOS ONE’s publication criteria as it currently stands. Therefore, we invite you to submit a revised version of the manuscript that addresses the points raised during the review process.

Please submit your revised manuscript by the 31st of March. If you will need more time than this to complete your revisions, please reply to this message or contact the journal office at plosone@plos.org. We apologise for the long delay in making a decision, it was mainly due to difficulties recruiting reviewers and conflicting suggestions.

Please include the following items when submitting your revised manuscript:A rebuttal letter that responds to each point raised by the academic editor and reviewer(s). You should upload this letter as a separate file labeled 'Response to Reviewers'.A marked-up copy of your manuscript that highlights changes made to the original version. You should upload this as a separate file labeled 'Revised Manuscript with Track Changes'.An unmarked version of your revised paper without tracked changes. You should upload this as a separate file labeled 'Manuscript'.

We look forward to receiving your revised manuscript.

Kind regards,

Roza Chaireti

Academic Editor

PLOS ONE

Journal Requirements:

Reviewers' comments:

Reviewer's Responses to Questions

**Comments to the Author**

1. Is the manuscript technically sound, and do the data support the conclusions?

Reviewer #1: Yes

Reviewer #2: Yes

2. Has the statistical analysis been performed appropriately and rigorously? 

Reviewer #1: Yes

Reviewer #2: Yes

3. Have the authors made all data underlying the findings in their manuscript fully available?

Reviewer #1: Yes

Reviewer #2: Yes

4. Is the manuscript presented in an intelligible fashion and written in standard English?

Reviewer #1: Yes

Reviewer #2: Yes

5. Review Comments to the Author

Reviewer #1: In this study authors aimed to prospectively validate the

EPIPHANY Index and Hestia criteria in a broad sample of patients with cancer, with a view to their possible appropriateness in decision making.

With this purpose, authors made a prospective, multicenter study with nine hundred patients diagnosed with PE between 2017/2020 were recruited

The rate of serious complications at 15 days was 11.8% (9.8-14.1%). Of the

EPIPHANY low-risk patients, 2.4% (0.8-4.6%) had serious complications, as did 5.5% (2.9-8.7%) of the moderate-risk participants and 21.0% (17.0-24.0%) of those

with high-risk episodes.

The concept of the study is interesting although there are several limitations that author need to address:

Please, edit English and review the text to avoid misunderstandings.

Abstract:

Introduction of the abstract: please, verify this sentence “This study has validated the EPIHANY Index”.

include month in the period of inclusion

this sentence in the abstract need to be supported with data: “The EPIPHANY Index correlated with overall survival”

Introduction. More references should be included in the 30 day mortality secondary to PE to provide a global idea to readers. There are other studies with lower mortality at 30 days.

Methods section

- If investigators followed patients at 30 days, why they analyzed complications at 15 days?

- Authors need to specify in which moment patients were recruited. At diagnosis? How long from diagnosis PE?

- Objective: Secondary objective need to be clarified. Compare with “other models” what means? Due to it was a prospective study this objectives needed to be pre-specified. The other models need to be clear and named.

- Methods section: the outcome were complications or serious complications? This point is critical. Moreover, I did not find in the Methods sections a definition of complications or serious complications.

Results section

- Please use lung cancer instead of broncho-pulmonary cancer.

- Of note, 12.3% of the sample (110 patients) had PE during anticoagulant treatment. Patients with cancer that develop PE during anticoagulant treatment should not be included in the analyses. This population differ too much with patients with cancer that develop PE without anticoagulant treatment.

Table 2 should be revisited.

- File of any complication need to be the sum of the other files

- define other serious complications due to is critical to define the outcome.

- about RV failure. It is different from RV dysfunction? Please make a definition

- About Hypotension: This complication can be at the moment of the PE event and for that reason can be a marker of the severity more than a complication during hospitalization. The same for thrombolytic treatment.

Reviewer #2: Using a prospective approach, the authors evaluated serious complications of pulmonary embolisms among cancer patients and validated EPIPHANY index in this cohort. In this multi-institution study, 900 patients were recruited, and the rates of serious complications were reported. Further, the EPIPHANY Index was examined for these patients along with the Hestia criteria and their usefulness was assessed. The study is well-designed. However, there are a few issues that need to be clarified:

1. Introduction (Page 6: “Nonetheless, outpatient management of cancer-related PE is not based on quality evidence, and there is a glaring dearth of prospective data”): there is a good consensus in the guidelines regarding outpatient management of PE in cancer patients (https://doi.org/10.1016/S1470-2045(19)30336-5; https://doi.org/10.1200/JCO.19.01461;
https://doi.org/10.6004/jnccn.2018.0084). I suggest the authors change this statement.

2. Introduction (Page 6: “or the like”): Please list the rules that the authors are referring one.

3. Introduction (Page 7: “The EPIPHANY Index was designed as a modified extension of the Hestia clinical decision rule to pragmatically select patients with low-risk, cancer-related PE [26].”). Please make sure the reference is correct to this statement.

4. Inclusion criteria included active solid neoplasm. This needs to be stated in the title and the abstract, that the study is on cancer patients with solid tumors.

5. What is the timeframe for the “Anticoagulation therapy at diagnosis of PE”? Why did only 12.3% got anticoagulants at the time of diagnosis?

6. In table 1, please separate the type of PE. Maybe to Laterality, location, and presentation.

7. The type of care doesn’t add up to 900. What is the remaining group?

8. Some complications in Table 2 are not defined in the methods, please clarify if only the outcomes mentioned in the methods are considered serious complications within 15 days and were used in the validation step. If that’s the case, only keep these complications in table 2 and the remaining complications can be summarized in a supplemental table.

9. Please define the abbreviations in the manuscript on their first occurrence. Abbreviations in the tables need to be listed in the footnote of the tables as well.

10. Please use the full-scale for the y-axis in Annex figure 3

6. PLOS authors have the option to publish the peer review history of their article (what does this mean?). If published, this will include your full peer review and any attached files.

Reviewer #1: No

Reviewer #2: No

---

## [Author Response · Author response to Decision Letter 0]

7 Mar 2023

Firstly, we would like to express our gratitude to the editors of PLOS ONE for their perseverance in finding reviewers for this manuscript and for considering it as a meritorious research. Next, we respond point by point (in the file annexed to the submission) to each of the queries and concerns raised during the review process, incorporating the changes into the manuscript.

---

## [Editor Report · Decision Letter 1]

12 Apr 2023

Prediction of serious complications in patients with pulmonary thromboembolism and solid cancer: validation of the EPIPHANY Index in a prospective cohort of patients from the PERSEO Study

PONE-D-22-07801R1

Dear Dr. Sánchez-Cánovas, 

We’re pleased to inform you that your manuscript has been judged scientifically suitable for publication and will be formally accepted for publication once it meets all outstanding technical requirements. We apologise for the delays during the review process but it is paramount to ensure a thorough review. 

Kind regards,

Roza Chaireti

Academic Editor

PLOS ONE
---

## [Editor Report · Acceptance letter]

27 Apr 2023

PONE-D-22-07801R1 

Prediction of serious complications in patients with pulmonary thromboembolism and solid cancer: validation of the EPIPHANY Index in a prospective cohort of patients from the PERSEO Study 

Dear Dr. Sánchez Cánovas:

I'm pleased to inform you that your manuscript has been deemed suitable for publication in PLOS ONE. Congratulations! Your manuscript is now with our production department. 

Kind regards, 

on behalf of

Dr. Roza Chaireti 

Academic Editor

PLOS ONE